# NLRC5 Deficiency Reduces LPS-Induced Microglial Activation via Inhibition of NF-κB Signaling and Ameliorates Mice’s Depressive-like Behavior

**DOI:** 10.3390/ijms241713265

**Published:** 2023-08-26

**Authors:** Chen Sun, Yuqing Shen, Piaopiao Liu, Yi Shen, Yue Hu, Ping Li, Ying Zhang, Fengqin Miao, Jianqiong Zhang

**Affiliations:** 1Jiangsu Provincial Key Laboratory of Critical Care Medicine, Department of Microbiology and Immunology, School of Medicine, Southeast University, Nanjing 210009, China; chensun_2019@163.com (C.S.); yuqingshenseu@hotmail.com (Y.S.); zyx_0426@aliyun.com (Y.Z.); 2Key Laboratory of Developmental Genes and Human Disease, Ministry of Education, Southeast University, Nanjing 210009, China; i17356152863@163.com (P.L.); yishenseu@163.com (Y.S.); 15005185959@163.com (Y.H.); li_p@nfls.com.cn (P.L.); 101006402@seu.edu.cn (F.M.)

**Keywords:** NLRC5, microglia, NF-κB, neuroinflammation, depressive mouse model

## Abstract

Microglia are believed to be the key immune effectors of the central immune microenvironment, and their dysregulation is associated with neuroinflammation and mood disorders. Nucleotide-binding oligomerization domain-like receptor family caspase recruitment domain-containing five (NLRC5) is a new member of the Nod-like receptor family. Recently, NLRC5 has been reported to be expressed by microglia. Nonetheless, the exact roles of NLRC5 in microglial activation and its function in depression have not been investigated yet. Herein, we found that reducing NLRC5 decreased lipopolysaccharide (LPS)-induced secretion of pro-inflammatory cytokines (IL-1β, IL-6, and TNF-α) in primary cultured microglia and microglial cell lines but not in bone marrow-derived macrophages (BMDMs). In more detail, reducing NLRC5 diminished the secretion of LPS-induced cytokines by attenuating IKKα/β phosphorylation and inhibiting NF-κB signaling. Moreover, the expression of *Nlrc5* in the hippocampus of LPS- or chronic unpredictable mild stress (CUMS)-induced depressive mice was increased. In line with the in vitro findings, *Nlrc5* deficiency inhibited microglial activation in the mouse hippocampus and improved LPS- or CUMS-induced depressive-like behaviors. In summary, we demonstrated the critical role of NLRC5 in LPS-induced microglial activation and LPS- or CUMS-induced depressive mouse models.

## 1. Introduction

As a category of brain-resident immune cells, microglia maintain brain function by modulating the homeostasis of the intracerebral microenvironment [1]. Microglial activation acts as a beneficial modulator during brain infections [2]. However, abnormal activation of microglia has been shown in several neurodegenerative and mood disorders [3,4,5,6]. Microglia can identify and respond to endogenous or exogenous signals by expressing pattern recognition receptors (PRRs), which comprise Toll-like receptors (TLRs), Nod-like receptors (NLRs), RIG-I-like receptors (RLRs), and C-type lectin receptors (CLRs) [7,8]. The binding of PRRs to their ligands can activate the nuclear factor kappa-light-chain-enhancer of activated B cells (NF-κB) signaling, thus initiating the expression of many genes associated with neuroinflammation [9].

After being first cloned and characterized by Cui et al. in 2010, NLRC5, the largest member of the NLRs family, was found to be widely expressed in multiple kinds of cells and involved in several inflammatory diseases [7,8,10]. The regulation of the NF-κB pathway via NLRC5 in the peripheral inflammatory response is a cell-type-specific and context-dependent process. Overexpression of NLRC5 inhibits LPS-induced activation of the NF-κB pathway in alveolar macrophages [11]. Knockdown of NLRC5 promotes NF-κB activation upon TNF-α treatment in hepatic stellate cells [12]. On the contrary, NLRC5 supports the activation of the NF-κB pathway induced by high glucose treatment in peritoneal macrophages and mesangial cells [13]. Until recently, the expression and function of NLRC5 in the central nervous system have not been fully clarified. NLRC5 is found to be expressed in neurons and microglia but rarely in astrocytes in developing and adult mouse brains [14,15]. NLRC5 is also expressed by BV2 cells, a classical microglial cell line [14]. As reported in a recent study, NLRC5 promotes MPP^+^/LPS-induced microglial and astrocyte activation in Parkinson’s Disease (PD) models [16]. However, the regulation of NF-κB by NLRC5 in microglial cells upon different stimulations in other disease models remains unclear.

Major depressive disorder (MDD) is a complex disorder with dysfunction in multiple cell types in the brain. Microglia play a critical role in depression since they can sense stress and contribute to the establishment of a depressed state [17,18,19]. For example, psychosocial stress activates microglia and facilitates the secretion of IL-1β and TNF-α via an ATP-dependent pathway [20]. Thus, using anti-inflammatory drugs to manipulate the status of microglia becomes an attractive approach to the treatment of MDD [21]. Although a transcriptomic study of patients with psychiatric disorders reveals that NLRC5 is highly expressed in brain tissue [22], the exact function of NLRC5 in the regulation of microglia in the pathology of depression has not been investigated.

In this study, first, we explored the roles of NLRC5 in regulating LPS-induced activation of NF-κB signaling in primary cultured microglia, which was compared with its function in LPS-treated BMDMs. Then, we investigated the effect of *Nlrc5* knockdown on microglia activation and depressive-like behaviors in the LPS- or chronic unpredictable mild stress (CUMS)-induced depressive mice model. Our results demonstrated that NLRC5 positively regulated the expression of pro-inflammatory cytokines through activation of the NF-κB pathway in LPS-stimulated microglial cells but not in BMDMs. Consistent with in vitro findings, NLRC5 also promoted microglial activation and enhanced LPS- or CUMS-induced depressive-like behaviors in mice.

## 2. Results

### 2.1. Nlrc5 Deficiency Inhibited the Expression of Pro-Inflammatory Cytokines under LPS Stimulation in Primary Cultured Microglia but Not in BMDMs

To clarify the exact mechanism of NLRC5 in the regulation of microglial activation under LPS stimulation, microglia from brain tissues of WT or *Nlrc5*^−/−^ mice were cultured and stimulated with 10 ng/mL or 100 ng/mL LPS (Figure 1A). The purity of microglia was assessed by Iba-1 positive staining (Appendix A). The result indicated that *Nlrc5* deficiency in microglial cells significantly reduced the increased expression in *Il-1b*, *Il-6*, and *Tnf-a* mRNA after LPS treatment for 8 h (Appendix A) and 24 h in a dose-dependent manner (Figure 1B–D). Consequently, the secretion of IL-1β was also reduced when *Nlrc5*^−/−^ microglia were treated with LPS (Appendix A and Figure 1E). Knockdown of *Nlrc5* in BV2, a microglial cell line, by lentivirus carrying sh*Nlrc5* (LV-Nlrc5-RNAi) also greatly decreased the expression of *Il-1b*, *Il-6*, and *Tnf-a* and reduced the secretion of IL-1β induced by LPS stimulation, which was consistent with the results obtained from primary microglia (Appendix A).

As reported in previous studies, the knockdown of *Nlrc5* enhanced cytokine gene expression in a murine macrophage cell line, RAW264.7 [10]. Therefore, we evaluated the effect of *Nlrc5* deficiency on LPS-induced proinflammatory cytokine production in BMDMs from WT or *Nlrc5*^−/−^ mice (Figure 1F). In contrast to the results obtained from microglia, the absence of *Nlrc5* in macrophages enhanced or sustained the induced expression of *Il-1b*, *Il-6*, and *Tnf-a* mRNA and secreted IL-1β protein under the stimulation of LPS for 8 h (Appendix A) and 24 h (Figure 1G–J). Taken together, all these results suggest that NLRC5 regulates proinflammatory responses to LPS stimulation in different ways in microglia and BMDMs.

### 2.2. NLRC5 Regulates Microglial Activation via the NF-κB Signaling Pathway

Recently, NLRC5 has been verified to affect the secretion of pro-inflammatory cytokines through NF-κB or inflammasome pathways [8,23]. However, it remains to be identified whether NLRC5 also affects LPS-induced activation of microglia through the NF-κB signaling pathway. The BV2-shCtrl cells and BV2-sh*Nlrc5* cells were transfected with an NF-κB inducible luciferase reporter plasmid and treated with 10 ng/mL or 100 ng/mL LPS. The results revealed that the elevated luciferase activity by LPS stimulation was significantly inhibited in BV2-sh*Nlrc5* cells compared with BV2-shCtrl cells (Figure 2A). The activation of the NF-κB pathway was dependent on the nuclear translocation of p65 (RelA), and it was inhibited in BV2-sh*Nlrc5* cells (Figure 2B,C). This suggested that NLRC5 promoted NF-κB signaling in BV2 cells in response to LPS treatment.

The secretion of IL-1β is tightly controlled by dual signaling. It is initially transcribed to produce Pro-IL-1β and then cleaved by caspase-1 to secrete biologically active IL-1β [24]. Given that decreased secretion of IL-1β was detected in LPS-treated BV2-sh*Nlrc5* cells, we checked whether the inflammasome pathway was also affected by the knockdown of *Nlrc5*. The results showed that a lower level of NLRC5 did not affect the formation of P20 (cleaved caspase 1) in BV2-sh*Nlrc5* cells after LPS plus ATP treatment (Figure 2D,E), indicating that the cleavage of Pro-IL-1β by caspase 1 was not changed. The reduced secretion of IL-1β was dependent on inhibited transcription of pro-IL-1β by knockdown of *Nlrc5*.

### 2.3. NLRC5 Regulates Microglial Activation by Promoting the Phosphorylation of IKKα/β

To further clarify the upstream molecules of NF-κB that were regulated by NLRC5, the LPS-induced phosphorylation of IKKα/β and IκB was detected by Western blotting (WB). As shown in Figure 3A–C (10 ng LPS treatment) and Appendix A (100 ng LPS treatment), LPS-induced IKKα/β and IκB phosphorylation were attenuated in BV2-sh*Nlrc5* cells compared with that in BV2-shCtrl cells. Meanwhile, LPS-induced phosphorylation of TAK1, an upstream kinase of IKKα/β, was not affected by the knockdown of *Nlrc5* (Figure 3D,E). This indicates that NLRC5 might regulate the NF-kB pathway by phosphorylating IKKα/β. NLRC5 antibodies were used to immunoprecipitate IKKα/β. Both phosphorylated and total IKKα/β were detected by immunoblot. As shown in Figure 3F, LPS increased the phosphorylated IKKα/β bound by NLRC5 in a dose-dependent manner. The knockdown of *Nlrc5* reduced the binding of NLRC5 to both phosphorylated and total IKKα/β proteins. In conclusion, these results revealed that NLRC5 phosphorylated IKKα/β, thus promoting the activation of NF-κB signaling in the LPS-treated microglial cells.

Contrary to the results of microglia, our findings revealed that the knockdown of *Nlrc5* in RAW264.7 cells did not decrease the phosphorylation of IKKα/β and IκB at all the time points we checked after LPS treatment (Figure 3G–I, 10 ng LPS treatment, and Appendix A, 100 ng LPS treatment), confirming that NLRC5 differently regulated LPS-induced IKKα/β phosphorylation in microglia and macrophages.

### 2.4. Nlrc5 Deficiency Ameliorated LPS- or CUMS-Induced Depressive and Anxiety-like Behaviors in Mice

Considering that the abnormal activation of microglia was shown to be an important factor in mood disorders, further exploration was conducted to identify whether NLRC5-related microglia activation played a critical role in depression. Therefore, LPS was intraperitoneally injected into WT or *Nlrc5* knockout (*Nlrc5*^−/−^) mice for five consecutive days to establish a mouse depression model, and some behavioral experiments, including OFT, SPT, TST, and FST, were performed (Figure 4A). We also established another depression model by exposing mice to chronic and unpredictable stress, including food/water deprivation, inversion light/dark cycles, and hot/cold stimulation for 35 days. Subsequently, corresponding behavioral experiments were also conducted (Figure 4H). First, an investigation was performed to identify whether the *Nlrc5* expression was changed in the LPS or CUMS models. The hippocampus tissue of mice was collected, and the mRNA level of *Nlrc5* was detected by RT-qPCR. Compared with the untreated group, the stimulation of LPS and CUMS increased *Nlrc5* expression in the hippocampal region (Appendix A).

Then, we determined the effect of *Nlrc5* knockout on LPS-induced and CUMS-induced depressive-like behaviors. The total travel distance in OFT was similar in WT and *Nlrc5*^−/−^ mice (Appendix A). The depletion of *Nlrc5* increased the frequency with which mice entered the central region and extended the time they stayed there compared with WT mice treated with LPS in OFT (Figure 4B–D). Besides that, the knockout of *Nlrc5* also ameliorated the decreased sucrose preference under LPS stimulation in SPT (Figure 4E). Meanwhile, the increase in immobility time in FST caused by LPS was reversed in *Nlrc5*^−/−^ mice (Figure 4G). In TST, *Nlrc5*^−/−^ mice showed a similar trend toward the relief of depression; however, the behavioral changes did not reach statistical significance (Figure 4F). All these results collectively showed that the depletion of *Nlrc5* could significantly alleviate LPS-induced depressive-like behaviors. In line with this, the knockout of *Nlrc5* also alleviated depressive behaviors in the CUMS-induced depressive model as well as in the LPS model (Figure 4I–N). Taken together, these animal behavior data suggested that *Nlrc5* deficiency ameliorated depressive behaviors in mice under different stress patterns.

### 2.5. Nlrc5 Deficiency Inhibited Microglial Activation Induced by LPS or CUMS

Next, we compared the expression of several pro-inflammatory genes in the hippocampal region of WT and *Nlrc5*^−/−^ mice after LPS treatment. As shown in Figure 5A–C, the expression of *Il-1b* and *Tnf-a* rather than *IL-6* was elevated in wild-type mice after 5 days of LPS treatment. However, these effects were significantly attenuated in *Nlrc5*^−/−^ mice. Consistent with this finding, all these pro-inflammatory genes were also upregulated in the hippocampus of CUMS-induced depressive mice, with the increased expression being attenuated by *Nlrc5* knockout (Figure 5F–H). Since the activation of microglia might change their numbers [25], immunohistochemical staining was performed on the hippocampal tissues. Compared with the control group, the number of Iba-1 positive microglia was increased in the hippocampus of LPS-treated mice. However, in the absence of *Nlrc5*, these changes were almost abrogated (Figure 5E). In the hippocampus of CUMS-induced depressive mice, the number of microglia was not changed either in WT or *Nlrc5*^−/−^ mice (Figure 5J). The quantitative analysis results in Figure 5E,J was shown in Figure 5D,I, respectively. Collectively, *Nlrc5* deficiency decreased the activation of microglia in the hippocampus of LPS- or CUMS-induced depressive mice.

## 3. Discussion

In this study, we demonstrated that NLRC5 promoted LPS-induced activation of canonical NF-κB signaling by forming polymers with IKKα/β and promoting IKKα/β phosphorylation, in turn leading to the phosphorylation of IκB and dissociation of the P50-P65 dimer. The P50-P65 dimer entered the nucleus and bound to the promoter of several proinflammatory cytokine genes, thus increasing their expression. Furthermore, the increased secretion of IL-1β was associated with the enhanced transcription of the *Il-1b* gene instead of the accelerated cleavage of pro-IL-1β. These findings revealed the function of NLRC5 in promoting the LPS-induced activation of microglia through the NF-κB signaling pathway (Figure 6). In line with the in vitro study, we found that *Nlrc5* deficiency inhibited microglial activation and attenuated depression-like behavior in both LPS- and CUMS-induced depressive mouse models.

NLRC5 is a multiple-functional immunomodulatory molecule. The role of NLRC5 in immune cells was first described as a molecular switch in the type I interferon response [26]. Moreover, NLRC5 is reported to be involved in the regulation of MHC class I transcription, inflammasome activation, and NF-κB signaling [26,27]. Enormous endeavors have been made to understand the regulation of the NF-κB signaling pathway by NLRC5. However, the results are controversial. Most of these reports reveal that NLRC5 is a negative regulator of NF-κB signaling [10,27,28]. In contrast to that, our results showed that NLRC5 promoted NF-κB activation in microglia. NLRC5 deletion does not affect the activation of NF-κB signaling in some cell types [27] or may positively regulate NF-κB signaling under certain circumstances [11]. Therefore, the role of NLRC5 in regulating the NF-κB signaling pathway depends on the cell type and stimulation patterns.

As the predominant resident immune cells in the brain, research on the expression and function of NLRC5 in microglia is scarce, and the results are controversial. HIV-1 Tat protein decreases NLRC5, which negatively regulates NF-κB activation in microglia [29]. However, in an MPP^+^/LPS-induced cellular PD model, NLRC5 positively regulates NF-κB activation in microglia [16]. Similarly, we found that NLRC5 promoted the activation of NF-κB signaling by LPS stimulation in microglia. This discrepancy may be explained by the different roles of NLRC5 in microglia responding to diverse stimulation patterns, which should be clarified through further investigation. Microglia are the resident macrophages in the central nervous system. However, the origins of microglia are different from those of BMDMs, which may correlate with their distinct reaction to stimulation [30]. We confirm this hypothesis by the fact that NLRC5 regulates NF-κB signaling differently in microglia and macrophages. Cui J et al. first described that the inhibitory role of NLRC5 was dependent on a direct interaction between NLRC5 and IKKα/β. They suggested that NLRC5 inhibited IKKα/β phosphorylation and kinase activity by competitively binding IKKα/β with NEMO [10]. They further reported that the strength of NLRC5 on NF-κB inhibition was determined by the cellular level of ubiquitinated NLRC5 in various kinds of macrophages [31]. According to our results, the loss of *Nlrc5* decreased its binding to both IKKα/β and phosphorylated IKKα/β, suggesting that NLRC5 provided a platform for IKKα/β phosphorylation in microglia. In the future, the different regulation patterns of NLRC5 on NF-κB activation in microglia and macrophages need to be further investigated.

Recently, a transcriptomic study on psychiatric diseases revealed that *Nlrc5* was highly expressed in the brains of patients [22], and we also found increased *Nlrc5* expression in the hippocampal region of depressive mice. Given that NLRC5 promoted LPS-induced microglial activation in cultured microglia and that microglial activation was critical in the etiology of depression, we investigated the role of NLRC5 in two different depressive mouse models: Systemic injections of LPS mimic depression symptoms in patients with acute infections. LPS triggers a strong central inflammatory response via Toll-like receptor four on microglia and induces acute depressive-like behaviors [32,33]. CUMS is a chronic stress model that recapitulates certain phenotypes and some aspects of the molecular pathology of MDD. It can induce chronic stress on microglia, thus affecting their function in several intracellular brain regions, with the most pronounced changes in the hippocampus [34,35]. The stressor induced by CUMS is still under investigation, and Gao et al. reported that CUMS-induced microglial activation was associated with cell surface P2X7R, a surface molecule that responded to ATP signals [36]. Although the molecular mechanisms in the two models are different, we demonstrated that the knockout of *Nlrc5* exerted the same effect on inhibiting microglial activation and improving animal depressive-like behaviors in both models. This suggested that *Nlrc5* played the same role in the microglia when they responded to different stress patterns. Therefore, targeting NLRC5 might be promising for the future treatment of patients with depression.

Zhang et al. reported that microglia were activated in the hippocampus of the LPS and CUMS models by demonstrating enlarged soma and hyperamified branches [37]. We also found increased branch number and length of microglia in these two models, and loss of *Nlrc5* attenuated these effects. However, the morphology of microglia is always dynamic, and the “activated” microglia are characterized by an amoeboid morphology in some reports. Therefore, it is recommended to use more gene and protein markers instead of morphologies to assess the state of microglia [38]. In this study, we found that loss of *Nlrc5* inhibited LPS- and CUMS-induced increases in cytokine expression in the hippocampus. *Nlrc5* knockout also decreased the number of microglia in the LPS-induced depression model. A deeper analysis of microglial states in *Nlrc5^−/−^* mice upon LPS or CUMS treatment will be considered in future studies.

It was reported that NLRC5 had a direct effect on neuron survival through the activation of NF-κB and AKT signaling in an MPTP-induced PD mouse model [16]. NLRC5 was also reported to protect neurons from hepatic ischemia/reperfusion- induced death [39]. Although neuronal death was seldom reported, several studies revealed dysregulation of synaptic plasticity and maladapted neurocircuitry in depression [40,41]. NLRC5 is also expressed in neurons and is an important regulator of Major Histocompatibility Complex class I in the developing hippocampus [42]. However, whether NLRC5 regulates synaptic plasticity is still unknown. In this study, conventional *Nlrc5* knockout mice were used to assess the role of NLRC5 in regulating depressive-like behaviors in mice. A knockout of NLRC5 in neurons might also contribute to the pathophysiology of depression. Therefore, further studies are required to delve into the function of NLRC5 in neurons and microglia in the etiology of depression.

## 4. Materials and Methods

### 4.1. Animals

*Nlrc5*^−/−^ (Cat#T003321) and C57BL/6J (Wild type, WT) mice were purchased from Nanjing GemPharmatech Co., Ltd. (Nanjing, China), and knockout of the *Nlrc5* gene was confirmed previously [42]. All the mice were maintained under pathogen-free conditions in our animal facility. A specific survival environment temperature was maintained at 20 ± 2 °C, and a 12-h light/night cycle was provided. Unless otherwise stated, animals could receive normal food and water voluntarily. Only males were included in the experiments. All animal experiments were conducted in accordance with ethical guidelines and were approved by the Southeast University Laboratory Animal Committee (Approval ID: syxk-2010.4987). To minimize pain, the mice were anesthetized by intraperitoneal injection of 1% sodium pentobarbital (10 mL/kg) before sacrifice.

### 4.2. Cell Culture

Isolation and culture of microglia: Primary microglia were isolated from mice at postnatal day 0 (p0)~p3. Then, the meninges and blood vessels were excised from brain tissues, and single-cell suspensions were prepared using trypsin. Subsequently, these cells were cultured in Dulbecco’s modified Eagle’s medium (DMEM, Viva Cell Biosciences, Shanghai, China, C3110-0500), containing 10% fetal bovine serum (FBS, Gibco, Grand Island, NY, USA, 10099141) and 1% Penicillin-Streptomycin (PS, Gibco, 15070063). After 3 days, the medium was replaced with 1 ng/mL granulocyte-macrophage colony-stimulating factor (GM-CSF, Sigma, St. Louis, MO, USA, SRP3201) every 2 days. Starting on day 7, the suspension cells were collected from the culture, identified by Iba1 immunostaining, and used for subsequent experiments.

Isolation and culture of BMDM: The BMDMs were prepared as previously reported [43]. In short, the bone marrow was washed out of the mouse femurs and cultured in DMEM medium supplemented with 10% FBS and 1% PS. To obtain purified macrophages, 20 ng/mL of macrophage colony-stimulating factor (M-CSF, Novoprotein, Suzhou, China CB34) was added every 2 days. After 1 week, the cells were collected and identified by F4/80 and CD11b staining for subsequent experiments.

Establishment of BV2 cell lines with stable knockdown of NLRC5: The BV2 cell line was kindly provided by Professor Honghong Yao, Southeast University, China. We constructed the BV2 cell line with stable knockdown of NLRC5 (BV2-sh*Nlrc5*) and the control cell line (shCtrl) by infection with LV-*Nlrc5*-RNAi and LV-scrambled-RNAi lentivirus, respectively (Shanghai Jikai Gene Technology Co., Ltd., Shanghai, China). After infection with lentivirus, the cells were diluted to contain only one cell per well and incubated until sufficient cells of a single clone were obtained. The knockdown of NLRC5 was verified by WB analysis.

Culture of RAW264.7 cells: The RAW264.7 cell line was purchased from Procell Life Science and Technology Co., Ltd. (Procell CL-0190, Wuhan, China) and cultured in DMEM medium supplemented with 10% FBS and 1% PS.

### 4.3. Western Blotting

WB was performed as per our previous protocol [42]. The corresponding antibodies to the protein of interest for immunoblotting included p-IKKα/β (Cell Signaling Technology, Danvers, MA, USA, 2697, 1:1000 dilution), p-IkB (Cell Signaling Technology, 9246, 1:1000 dilution), IKKα/β (Abcam, Cambridge, MA, USA, ab178870, 1:1000 dilution), IkB (Cell Signaling Technology, 9242, 1:1000 dilution), p65 (BD Bioscience, San Jose, CA, USA, 610868, 1:1000 dilution), NLRC5 (Abcam, ab105411, 1:500 dilution), p-TAK1 (ImmunoWay, Plano, TX, USA, YP1522, 1:500 dilution), TAK1 (ImmunoWay, YT4536, 1:1000 dilution), cleaved-Caspase-1 p20 (Santa Cruz Biotechnology, Santa Cruz, CA, USA, sc1218, 1:1000 dilution), β-actin (Sigma-Aldrich, A5441, 1:5000 dilution), and Histone-3 (Cell Signaling Technology, 4499, 1:2000 dilution). The WB band was analyzed and quantitated by ImageJ 1.6.0 (NIH, Bethesda, MD, USA).

### 4.4. Real-Time Quantitative PCR (RT-qPCR)

Total RNA from mouse tissues or cells was obtained using Trizol reagent (Sigma, T9424) according to the product instructions. Subsequently, cDNA generation and gene expression were determined as previously described [42]. The sequences of all the primers are listed in Appendix A. The relative expression of mRNA was calculated using the 2^−∆∆CT^ method. The mRNA content of β-actin was used as the internal control.

### 4.5. Enzyme-Linked Immunosorbent Assay (ELISA)

The debris in the cell culture supernatant was removed by centrifugation at 4 °C, 1000× *g*, for 10 min. Then, the secreted IL-1β in the supernatants was measured by the ELISA kit (R&D systems, Minneapolis, MN, USA, MLB00C) according to the commercial instructions.

### 4.6. Plasmid Transfection and Luciferase Reporter Assay

The activation of NF-κB was measured by the luciferase reporter assay. In short, cells were co-transfected with 1000 ng of the NF-κB reporter plasmid and 80 ng of the control plasmid (RL-TK, to quantify transfection efficiency). Both plasmids were kindly provided by Professor Chunguang Yan, Southeast University, China. Different concentrations of LPS were added to the cell cultures 48 h after transfection, and luciferase activity was measured using a dual luciferase reporter kit (Promega, E1910, Madison, WI, USA) by a TD 20/20n luminometer (Turner Biosystems, San Jose, CA, USA) 24 h later.

### 4.7. Immunoprecipitation

Total cellular proteins were extracted using a lysis buffer for immunoprecipitation. Protein concentrations were measured by the BCA Protein Assay Kit (Thermo Scientific, Waltham, MA, USA, 23227). Immunoprecipitation experiments were performed as previously described [42]. The magnetic beads-bound proteins were separated using SDS/PAGE and analyzed by WB.

### 4.8. Mouse Model

LPS-induced mouse model of depression: The LPS model of depression was constructed as previously reported [37]. The mice were intraperitoneally injected with LPS (1 mg/kg, Sigma, L2630) for five consecutive days. Then, behavioral experiments were performed. After that, hippocampus tissues were collected and immediately stored at −80 °C.

CUMS-induced mouse model of depression: The CUMS model of depression was constructed as per a validated protocol [37]. Briefly, mice were exposed to randomly scheduled low-intensity social and environmental stressors, including food/water deprivation, reversed light/dark cycles, and heat/cold stimuli, 2–3 times a day for 35 days. Subsequently, behavioral tests were performed, followed by systematic perfusion and the collection of hippocampus tissues.

### 4.9. Behavior Test

Open-field square test (OFT): Mice were habituated in the laboratory for 2 h and then introduced to the open-field apparatus (40 × 40 × 40 cm^3^). The video was recorded for 20 min. Then, the Ethovision XT system (Noldus Information Technology, wageningen, the Netherlands) was used to automatically calculate the total distance that mice traveled in the open field, the time they spent in it, or the number of times they entered the central square (20 × 20 cm^2^).

Sucrose preference test (SPT): Anhedonia is a typical symptom of depression, and the SPT is designed for relevant detection. Mice were given two bottles filled with pure water or 1% sucrose solution (100 mL for each). The bottle positions were changed every 12 h. After 24 h, the bottles were weighed again to calculate the consumption of pure water and sucrose solution. Sucrose preference (%) = sucrose consumption/total liquid consumption × 100%.

Tail suspension test (TST): Mice were suspended 60 cm above the floor by the tape, which was placed approximately 1 cm from the tip of the tail. Each test lasted 6 min, with the middle 4 min used to count the immobility time.

Forced swimming test (FST): Mice were individually placed in a cylinder (diameter: 20 cm) filled with 15–20 cm of water with a temperature maintained at 25 ± 1 °C. The time for mice to float on the water surface without active activity was defined as “immobility time”. In a 6-min test, the immobility time was recorded within the middle 4 min.

### 4.10. Immunostaining and Image Analysis

After anesthetization and perfusion with refrigerated 1× phosphate-buffered saline (PBS), the brains were cut out and fixed in 4% paraformaldehyde (Sigma Aldrich, 158127). They were then embedded in an optimal cutting temperature compound and refrigerated at −80 °C. Brain sections (30 μm thick) were prepared using a frozen sectioning machine (Leica, Wetzlar, Germany, CM1950). Then, they were rinsed in 1× PBS, incubated in H_2_O_2_ to remove peroxidase, and immersed in 10% normal goat serum to block the non-specific binding. At last, the sections were incubated with anti-Iba-1 antibody (Wako, Osaka, Japan, 019-19741, 1:500 dilution) overnight at 4 °C and with goat anti-rabbit antibody at 37 °C for 15–20 min. As a final step of the experiment, the Enhanced DAB Peroxidase Substrate Kit (Servicebio, Wuhan, China, G1212) was used to visualize Iba-1-positive cells. We randomly selected the fields of view from consecutive sections and counted the number of microglia using ImageJ 1.6.0.

### 4.11. Statistical Analysis

In this study, contingencies were eliminated by repeating the experiment at least three times. The statistical analyses between the two groups were performed using the Student’s *t*-test, and the comparisons among multiple groups were analyzed by the two-way analysis of variance (ANOVA) using GraphPad Prism 8.0.1. The number of replicates for each experiment was denoted as “n=”. *p* ≤ 0.05, which indicated that the data were statistically significant. Unless otherwise stated, no data were excluded from the analysis.

## Figures and Tables

**Figure 1 ijms-24-13265-f001:**
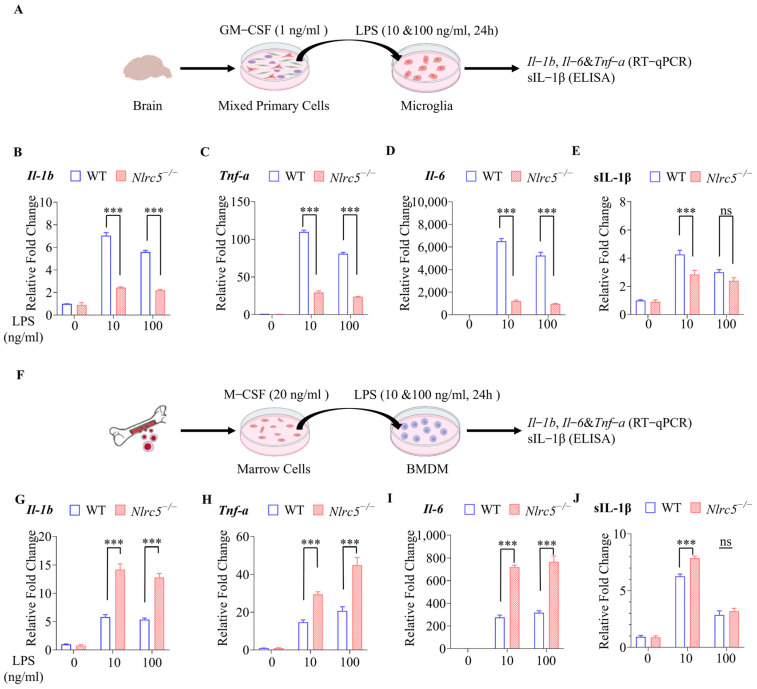
NLRC5 deficiency inhibited LPS-induced pro-inflammatory cytokine production in primary cultured microglia but not in bone marrow derived macrophages (BMDM). (**A**) Flow chart of primary microglia culture from WT or NLRC5^−/−^ mice. (**B**–**E**) Effect of NLRC5 knockout on cytokine production in primary cultured microglia. The mRNA (n = 3 for each group) of *Il-1b* (**B**), *Tnf-a* (**C**), and *Il-6* (**D**) of microglia treated with LPS or Saline for 24 h was detected by RT-qPCR, and secreted IL-1β (**E**) in the supernatant of primary microglia was detected by ELISA (n = 5 for each group). (**F**) Flow chart of Isolation of BMDM from WT or NLRC5^−/−^ mice. (**G**–**J**) Effect of NLRC5 knockout on cytokine production in BMDM. The mRNA (n = 3 for each group) of *Il-1b* (**G**), *Tnf-a* (**H**), and *Il-6* (**I**) in BMDM treated with LPS or Saline for 24 h was detected by RT-qPCR, and IL-1β secretion (**J**) in the supernatant of BMDM was detected by ELISA (n = 5 for each group). All data were presented as mean ± SEM ((**B**), LPS: F (2, 12) = 354.6, *p* < 0.001; NLRC5: F (1, 12) = 490.3, *p* < 0.001; Interaction: F (2, 12) = 121.6, *p* < 0.001; **C**, LPS: F (2, 12) = 1207, *p* < 0.001; NLRC5: F (1, 12) = 1493, *p* < 0.001; Interaction: F (2, 12) = 397.4, *p* < 0.001; (**D**), LPS: F (2, 12) = 357.4, *p* < 0.001; NLRC5: F (1, 12) = 651.9, *p* < 0.001; Interaction: F (2, 12) = 168.8, *p* < 0.001; **E**, LPS: F (2, 24) = 77.89, *p* < 0.001; NLRC5: F (1, 24) = 16.67, *p* < 0.001; Interaction: F (2, 24) = 4.976, *p* < 0.05; (**G**), LPS: F (2, 12) = 173.2, *p* < 0.001; NLRC5: F (1, 12) = 138.6, *p* < 0.001; Interaction: F (2, 12) = 38.21, *p* < 0.001; (**H**), LPS: F (2, 12) = 134.2, *p* < 0.001; NLRC5: F (1, 12) = 65.07, *p* < 0.001; Interaction: F (2, 12) = 19.13, *p* < 0.001; (**I**), LPS: F (2, 12) = 306.0, *p* < 0.001; NLRC5: F (1, 12) = 224.7, *p* < 0.001; Interaction: F (2, 12) = 56.22, *p* < 0.001; (**J**), LPS: F (2, 24) = 395.4, *p* < 0.001; NLRC5: F (1, 24) = 11.94, *p* < 0.01; Interaction: F (2, 24) = 7.405, *p* < 0.01). A two-way ANOVA was used to determine the statistical significance between the two groups. *** *p* ≤ 0.001. ns: no significant difference. GM-CSF: Granulocyte macrophage colony-stimulating factor; M-CSF: Macrophage colony-stimulating factor; BMDM: bone marrow-derived macrophages.

**Figure 2 ijms-24-13265-f002:**
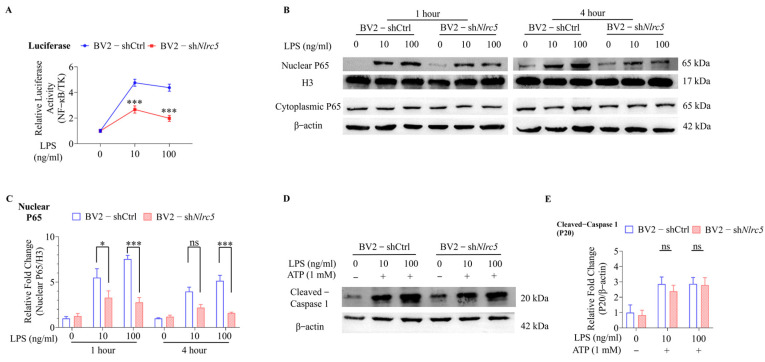
NLRC5 regulates microglial activation via the NF-κB signaling pathway in BV2 cells. (**A**) NF-κB dependent luciferase activity was measured in BV2-shCtrl and BV2-sh*Nlrc5* cells treated with different concentrations of LPS. (**B**) Representative Western blot image of nuclear/cytoplasmic P65 in BV2-shCtrl and BV2-sh*Nlrc5* cells treated with LPS for 1 h and 4 h, respectively. (**C**) The relative expression of nuclear p65 was normalized to that of nuclear H3 and shown as a ratio to that in untreated BV2-shCtrl cells (n = 5 for each group). (**D**) Representative Western blot image of cleaved-caspase 1 in BV2-shCtrl and BV2-sh*Nlrc5* cells treated with LPS plus ATP. (**E**) The relative expression of cleaved-caspase 1 in LPS-treated groups was normalized to that of β-actin and shown as a ratio to that in untreated BV2-shCtrl cells (n = 5 for each group). All data were presented as mean ± SEM ((**A**), LPS: F (2, 12) = 239.0, *p* < 0.001; NLRC5: F (1, 12) = 194.0, *p* < 0.001; Interaction: F (2, 12) = 49.16, *p* < 0.001; (**C**), LPS: F (5, 48) = 23.54, *p* < 0.001; NLRC5: F (1, 48) = 49.92, *p* < 0.001; Interaction: F (5, 48) = 8.539, *p* < 0.001; (**E**), LPS: F (2, 24) = 11.81, *p* < 0.001; NLRC5: F (1, 24) = 0.4426, *p* = 0.5122; Interaction: F (2, 24) = 0.1129, *p* = 0.8937). A two-way ANOVA was used to determine the statistical significance between the two groups. * *p* ≤ 0.05 and *** *p* ≤ 0.001. ns: no significant difference.

**Figure 3 ijms-24-13265-f003:**
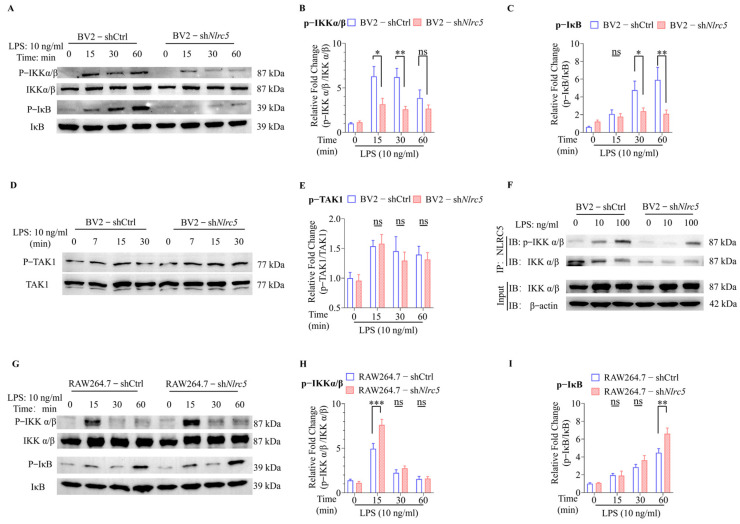
NLRC5 regulates microglial activation via promoting phosphorylation of IKKα/β. (**A**) Representative Western blot image of p-IKKα/β, IKKα/β, p-IκB and IκB in BV2-shCtrl and BV2-sh*Nlrc5* cells treated with LPS (10 ng/mL) for different times. Bars represent the relative fold change of p-IKKα/β to IKKα/β (**B**) and p-IκB to IκB (**C**) (n = 5 for each group). (**D**) Representative Western blot image of p-TAK1 and TAK1 in BV2-shCtrl and BV2-sh*Nlrc5* cells treated with LPS. (**E**) Bars represent the relative fold change of p-TAK1 to TAK1 (n = 5 for each group). (**F**) Anti-NLRC5 was used to precipitate IKKα/β from BV2-shCtrl and BV2-sh*Nlrc5* cell extracts. P-IKKα/β and IKKα/β were subsequently determined by Western blot analysis. (**G**) Representative Western blot image of p-IKKα/β, IKKα/β, p-IκB and IκB in RAW264.7-shCtrl and RAW264.7-sh*Nlrc5* cells treated with LPS. Bars represent the relative fold change of p-IKKα/β to IKKα/β. (**H**) and p-IκB to IκB (**I**) (n = 5 for each group). All data were presented as mean ± SEM ((**B**), LPS: F (3, 32) = 11.51, *p* < 0.001; NLRC5: F (1, 32) = 15.95, *p* < 0.001; Interaction: F (3, 32) = 3.201, *p* < 0.05; (**C**), LPS: F (3, 32) = 8.843, *p* < 0.001; NLRC5: F (1, 32) = 9.209, *p* < 0.01; Interaction: F (3, 32) = 4.286, *p* < 0.05; (**E**), LPS: F (3, 32) = 5.683, *p* < 0.01; NLRC5: F (1, 32) = 0.3470, *p* = 0.5600; Interaction: F (3, 32) = 0.1702, *p* = 0.9157; (**H**), LPS: (3, 32) = 78.14, *p* < 0.001; NLRC5: F (1, 32) = 8.092, *p* < 0.01; Interaction: F (3, 32) = 6.471, *p* < 0.01; (**I)**, LPS: (3, 32) = 47.36, *p* < 0.001; NLRC5: F (1, 32) = 6.760, *p* < 0.05; Interaction: F (3, 32) = 3.143, *p* < 0.05). A two-way ANOVA was used to determine the statistical significance of the two groups. * *p* ≤ 0.05, ** *p* ≤ 0.01, and *** *p* ≤ 0.001. ns: no significant difference. p-IKKα/β: phosphorylated IKKα/β; p-IκB: phosphorylated IκB; p-TAK1: phosphorylated TAK1.

**Figure 4 ijms-24-13265-f004:**
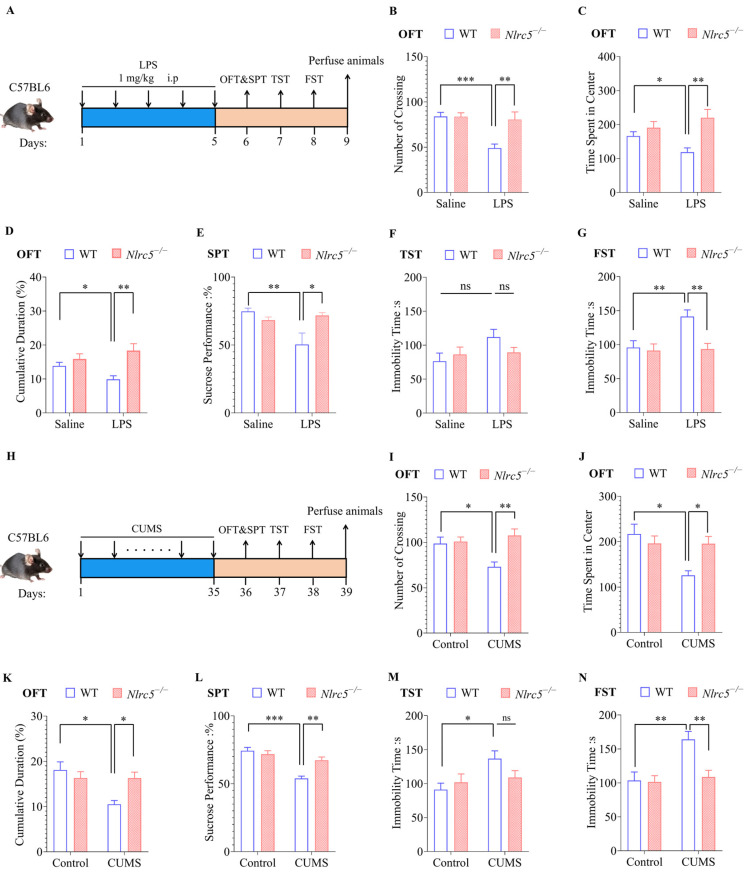
Knockout of NLRC5 ameliorates LPS- and CUMS-induced depression-like behavior. (**A**) Flow chart of establishing an LPS-induced depressive mouse model: WT and NLRC5^−/−^ mice were injected with LPS intraperitoneally (1 mg/kg) or saline for five consecutive days. Behavioral tests were then conducted, including OFT, SPT, TST, and FST. (**B**–**G**) Effect of NLRC5 knockout on depressive-like behavior in the LPS model (n = 12 for each group). (**H**) Flow chart of establishing a depressive mouse model induced by CUMS: mice were exposed to 2–3 stressors every day or were free of stressors for 35 consecutive days. Behavioral tests were then conducted. (**I**–**N**) Effect of NLRC5 knockout on depressive-like behavior in the LPS model (n = 12 for each group). All data were presented as mean ± SEM ((**B**), LPS: F (1, 44) = 11.01, *p* < 0.01; NLRC5: F (1, 44) = 7.334, *p* < 0.01; Interaction: F (1, 44) = 7.334, *p* < 0.01; (**C**), LPS: F (1, 44) = 12.88, *p* < 0.001; NLRC5: F (1, 44) = 0.2690, *p* = 0.6066; Interaction: F (1, 44) = 4.809, *p* < 0.05; (**D**), LPS: F (1, 44) = 12.88, *p* < 0.001; NLRC5: F (1, 44) = 0.2690, *p* = 0.6066; Interaction: F (1, 44) = 4.809, *p* < 0.05; (**E**), LPS: F (1, 44) = 5.015, *p* < 0.05; NLRC5: F (1, 44) = 2.540, *p* = 0.1182; Interaction: F (1, 44) = 8.967, *p* < 0.01; (**F**)**,** LPS: F (1, 44) = 3.309, *p* = 0.0757; NLRC5: F (1, 44) = 0.3489, *p* = 0.5578; Interaction: F (1, 44) = 2.334, *p* = 0.1337; (**G**), LPS: F (1, 44) = 6.252, *p* < 0.05; NLRC5: F (1, 44) = 7.584, *p* < 0.01; Interaction: F (1, 44) = 5.167, *p* < 0.05; (**I**), CUMS: F (1, 44) = 2.258, *p* = 0.1401; NLRC5: F (1, 44) = 8.674, *p* < 0.01; Interaction: F (1, 44) = 6.680, *p* < 0.05; (**J**), CUMS: F (1, 44) = 7.641, *p* < 0.01; NLRC5: F (1, 44) = 2.111, *p* = 0.1533; Interaction: F (1, 44) = 7.438, *p* < 0.01; (**K**), CUMS: F (1, 44) = 7.641, *p* < 0.01; NLRC5: F (1, 44) = 2.111, *p* = 0.1533; Interaction: F (1, 44) = 7.438, *p* < 0.01; (**L**), CUMS: F (1, 44) = 29.11, *p* < 0.001; NLRC5: F (1, 44) = 5.538, *p* < 0.05; Interaction: F (1, 44) = 11.89, *p* < 0.01; (**M**), CUMS: F (1, 44) = 3.059, *p* = 0.0873; NLRC5: F (1, 44) = 0.6049, *p* = 0.4409; Interaction: F (1, 44) = 5.767, *p* < 0.05; (**N**), CUMS: F (1, 44) = 9.561, *p* < 0.01; NLRC5: F (1, 44) = 6.867, *p* < 0.05; Interaction: F (1, 44) = 5.870, *p* < 0.05). A two-way ANOVA was used to determine the statistical significance between the two groups. * *p* ≤ 0.05, ** *p* ≤ 0.01, and *** *p* ≤ 0.001. ns: no significant difference. LPS: Lipopolysaccharide; CUMS: Chronic unpredictable mild stress; OFT: open field test; SPT: sugar preference test; TST: tail suspension test; FST: forced swimming test.

**Figure 5 ijms-24-13265-f005:**
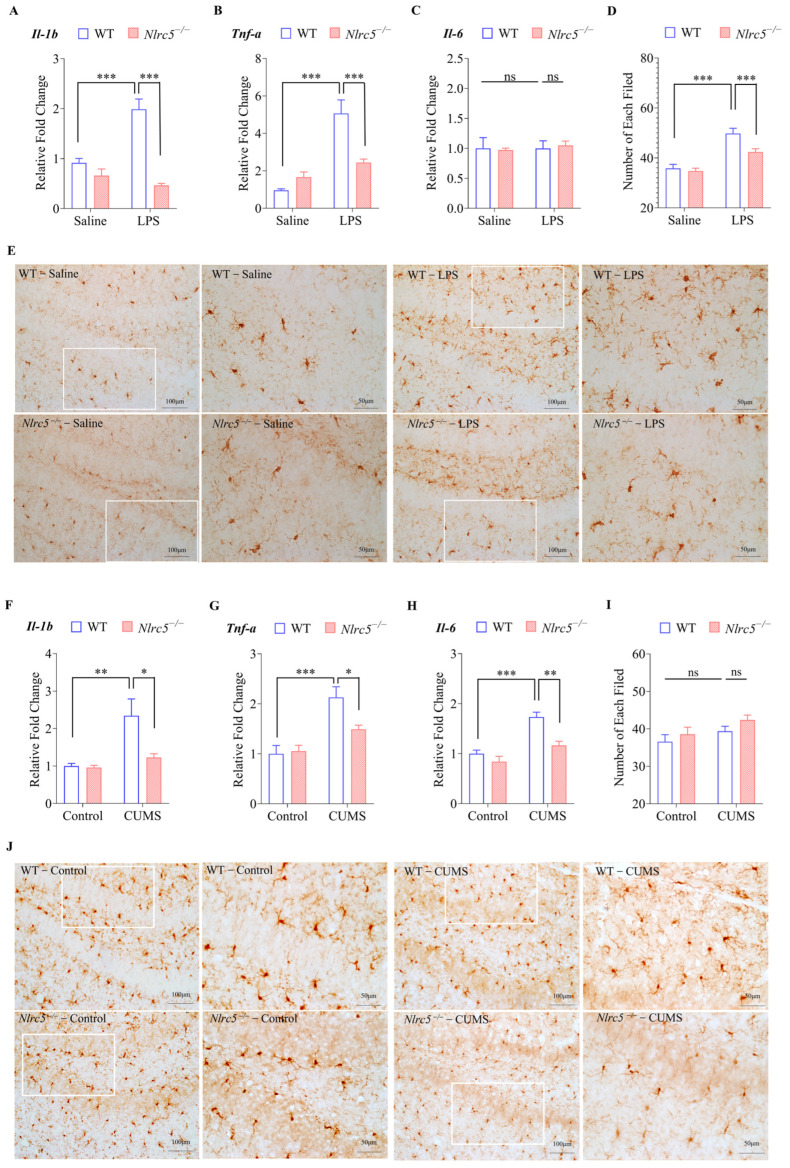
NLRC5 deficiency inhibits microglial activation induced by LPS or CUMS. (**A**–**C**,**F**–**H**) Effect of NLRC5 knockout on cytokine production in the hippocampal tissues of depression model mice induced by LPS or CUMS. The abundances of *Il-1b* (**A**,**F**), *Tnf-a* (**B**,**G**), and *Il-6* (**C**,**H**) were measured by RT-qPCR (n = 5 for each group). (**E**,**J**) Representative images of Iba-1-positive microglia in the hippocampal tissues of depression model mice induced by LPS or CUMS. An enlarged image of the area within the white box is shown on the right. ImageJ software 1.6.0 was used to analyze the total number of Iba-1-positive microglia in each field (**D**,**I**) (n = 30 cells from 4 mice for each group). All data were presented as mean ± SEM ((**A**), LPS: F (1, 16) = 11.16, *p* < 0.01; NLRC5: F (1, 16) = 46.04, *p* < 0.001; Interaction: F (1, 16) = 23.52, *p* < 0.001; (**B**), LPS: F (1, 16) = 40.05, *p* < 0.001; NLRC5: F (1, 16) = 6.736, *p* < 0.05; Interaction: F (1, 16) = 15.91, *p* < 0.01; (**C**), LPS: F (1, 16) = 0.1065, *p* = 0.7484; NLRC5: F (1, 16) = 0.008776, *p* = 0.9265; Interaction: F (1, 16) = 0.1133, *p* = 0.7408; (**D**), LPS: F (1, 116) = 46.37, *p* < 0.001; NLRC5: F (1, 116) = 7.169, *p* < 0.01; Interaction: F (1, 116) = 4.011, *p* < 0.05; (**F**), CUMS: F (1, 16) = 11.88; NLRC5:, *p* < 0.001; NLRC5: F (1, 16) = 6.102, *p* < 0.05; Interaction: F (1, 16) = 5.308, *p* < 0.05; (**G**), CUMS: F (1, 16) = 26.24, *p* < 0.001; NLRC5: F (1, 16) = 3.651, *p* = 0.0741; Interaction: F (1, 16) = 5.181, *p* < 0.05; (**H**), CUMS: F (1, 16) = 34.66, *p* < 0.001; NLRC5: F (1, 16) = 16.24, *p* < 0.01; Interaction: F (1, 16) = 5.091, *p* < 0.05; (**I**), CUMS: F (1, 116) = 4.159, *p* < 0.05; NLRC5: F (1, 116) = 2.387, *p* = 0.1251; Interaction: F (1, 116) = 0.09547, *p* = 0.7579). A two-way ANOVA was used to determine the statistical significance between the two groups. * *p* ≤ 0.05, ** *p* ≤ 0.01, and *** *p* ≤ 0.001. ns: no significant difference.

**Figure 6 ijms-24-13265-f006:**
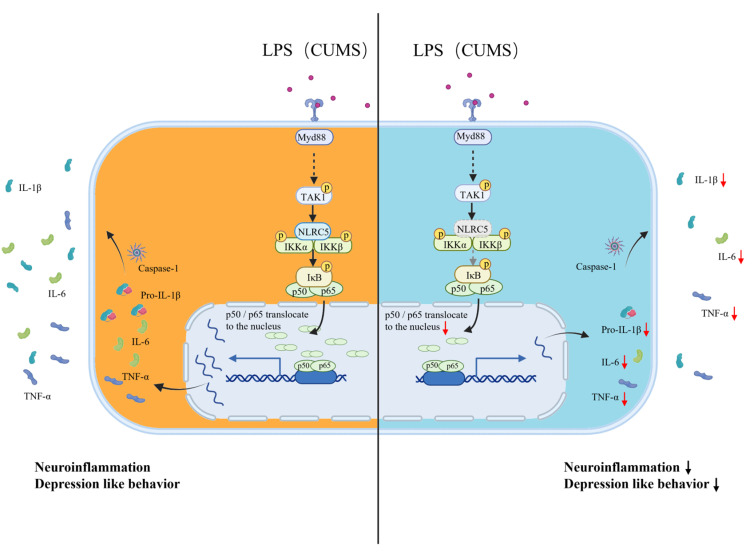
Schematic illustration of the function of NLRC5 in LPS-induced activation of microglia and production of pro-inflammatory cytokines. Black arrow: activation. Curved arrow: transport into or out of the cell nucleus or transport out of the cell. Dashed black arrow: interacting proteins not confirmed in this study. Dashed gray arrow: decreased activation. Dashed circle: loss of NLRC5. Red arrow: decreased expression.

## Data Availability

The data presented in this study are available on request from the corresponding author.

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
