# Peer review of "NLRC5 Deficiency Reduces LPS-Induced Microglial Activation via Inhibition of NF-κB Signaling and Ameliorates Mice’s Depressive-like Behavior"

_ijms, 2023, doi:10.3390/ijms241713265_

Round 1
Reviewer 1 Report
The authors elaborated a well-designed study to investigate the role of NLRC5 (and its knockout) in the context of depressive-like behavior. Authors showed that NLRC5 plays specific roles in microglia, that are not recapitulated by other macrophages, and explored canonical pathways known to be involved on the NLRC5 regulation, validating them for both LPS and CUMS stimuli. A very consistent set of data with the inhibition of NLRC5 regulating IKK and p50/p65 dynamics, inducing IL-1b gene expression and release, together with an adequate behavioral assessment. The English writing is very good but could be improved in terms of minor typos and repetitive sentence structure in the results section. Overall, I enjoyed reviewing this work, as I believe it can point NLRC5 as a promising therapeutic target to explore in depression and related areas. I also foresee that NLRC5 can be relevant to be explored in the context of neuroinflammation and associated-disorders, since potent effects were observed against LPS.
Please consider the following questions:
Q1. Introduction (line 45-51): This part is confusing. It is not clear weather NLRC5 increase/decrease can lead to the inhibition of LPS-induces NFKb activation in alveolar macrophages. Especially considering what comes next about the effects with the knockdown causing NFkB activation in hepatic stellate cells.
Besides, since some of this evidence is contrary to your findings, please discuss why is this in the discussion section.
Q2. Results (line 88): Excellent results with the knockdown. Nonetheless, it should be briefly described here which type of knockdown was made. Even if it comes in the methods in more detail.
Q3: Line 127: “As shown in Figure 3F, phosphorylating of NLRC5 bound IKKα/β were enhanced by LPS” This phrase is confusing to read.
Q4. Line 129-130: “In conclusion, these results revealed that NLRC5 bound to and phosphorylated IKKα/β,”. Again this is confusing. DO you mean “binds to phosphorylated IKKα/β, thus promoting the activation…”. It would help if you improve the language in these important results.
you should first state here that you immunoprecipitated IKKa/b with NLRC5 antibody. Only then you understand the "bound" in this section.
Q5. Line 148: “Compare”. It should be "Comparing" or “Compared”
Q6. Line 172-174: Microglia morphology can be related with their phenotype/function, but not just like that. Nowadays its important to consider many more functional parameters to access that microglial function. Please give a look on this article and be careful when describing microglia and defining activation (doi: 10.1016/j.neuron.2022.10.020).
Q7. Line 175-176: “….and their branches were more and longer indicating the activation of microglial cells.” This sentence is highly controversial. What means “activated” microglia? It can be a world of possible phenotypes. Besides, the most common “activated” microglia, usually associated with negative effects, is usually characterized by an amoeboid morphology, which is the opposite of what you describe here. The higher number of iba1 cells it’s the only result that is typical of activated microglia, but it may still depend on the type of activation… Usually, highly ramified microglia are traditionally associated to the (non-existent) “resting” state. It could be the case that your ramified microglia became unable to respond to the stimuli due to the NLRC5 knockout and its consequent inhibition of NFkB pathways. And this can be a relevant side effect, in my point of view! Since it was not your idea to explore microglia in this work, at least refer that a much deeper analysis of microglia should be considered in further studies.
Q8. Figure 4: I was impressed by how well the CUMS long-term effects recapitulate the LPS short effects. The results were incredibly similar. Please highlight and elucidate this topic in the discussion section. That may justify why the knockout worked so well on both stimuli.
Q9. Figure 6: In my opinion this figure is good, but you should also add the results that you found upon NLRC5 knockout and highlight them. You should use inhibition arrows pointing to NLRC5 and the beneficial outcomes that you demonstrated through this work.
The English is good, but can be improved in the results section.
Reviewer 2 Report
In this article the authors examined the consequences of the NLRC5 deficiency in LPS-induced microglial activation and compared this effect in LPS-treated bone marrow-derived macrophages . They concluded that NLRC5 positively regulated the expression of pro-inflammatory cytokines only in microglia but not in BMDM.
Apart from the fact that the rationale is not well described, as reference is made to a different effect on the two cell populations and secondarily an in vivo model of depressive-like behaviors in mice is considered, there are several unclear aspects that emerge from reading the manuscript
Line 63 page 2: In the introduction, please specify MMD with the full name. This is also valid for CUMS (line 71).
Figure2:
It does not seem to me that in figure 2 Panel D (cleaved caspase 1) there is the same match as in the original blots. In addition, there are problems with the significance: the graph (Panel C) shows SEM values with high significance (P<0.001) this is not apparent from the graph representation, so the authors need to revise the calculation of significances.
Figure 3
Equally in Figure 3 panel A and panel D and panel F: the images do not match the original.
Figure 5: the quality of the representative images of Iba-1-positive microglia in the hippocampal tissues of depression model mice induce by LPS or CUMS is very poor. In addition, it also lacks any graphic indication of Iba-1-positive microglial cells in each field besides the indication of branches
Finally, the authors should provide a more convincing justification regarding the different effect by NLRC on different cell types and not simplistically state that the differences are related to cell type. Microglia ultimately represent the macrophage population of the central nervous system.
A careful check of the English by a native speaker is need.
Round 2
Reviewer 2 Report
No comment
Author Response
We thanks for the reviewer to approve our manuscript.